# Identification of triple-negative breast cancer cell lines classified under the same molecular subtype using different molecular characterization techniques: Implications for translational research

Jose Rodrigo Espinosa Fernandez[1], Bedrich L. Eckhardt[1¤a], Jangsoon Lee[1], Bora Lim[1], Troy Pearson[1], Rob S. Seitz[2], David R. Hout[2], Brock L. Schweitzer[2], Tyler J. Nielsen[2], O. Rayne Lawrence[2], Ying Wang[3], Arvind Rao[3¤b], Naoto T. Ueno[1]*

1 Department of Breast Medical Oncology, The University of Texas MD Anderson Cancer Center, Houston, Texas, United States of America, 2 Insight Genetics Incorporated, Nashville, Tennessee, United Sates of America, 3 Department of Biostatistics, The University of Texas MD Anderson Cancer Center, Houston, Texas, United States of America

¤a Current address: Olivia Newton-John Cancer Research Institute and School of Cancer Medicine La Trobe University, Heidelberg, Australia
¤b Current address: Department of Computational Medicine & Bioinformatics, University of Michigan, Ann Arbor, Michigan, United States of America
* nueno@mdanderson.org

## Abstract

The original algorithm that classified triple-negative breast cancer (TNBC) into six subtypes has recently been revised. The revised algorithm (TNBCtype-IM) classifies TNBC into five subtypes and a modifier based on immunological (IM) signatures. The molecular signature may differ between cancer cells *in vitro* and their respective tumor xenografts. We identified cell lines with concordant molecular subtypes regardless of classification algorithm or analysis of cells *in vitro* or *in vivo*, to establish a panel of clinically relevant molecularly stable TNBC models for translational research. Gene expression data were used to classify TNBC cell lines using the original and the revised algorithms. Tumor xenografts were established from 17 cell lines and subjected to gene expression profiling with the original 2188-gene algorithm TNBCtype and the revised 101-gene algorithm TNBCtype-IM. A total of six cell lines (SUM149PT (BL2), HCC1806 (BL2), SUM149PT (BL2), BT549 (M), MDA-MB-453 (LAR), and HCC2157 (BL1)) maintained their subtype classification between *in vitro* and tumor xenograft analyses across both algorithms. For TNBC molecular classification–guided translational research, we recommend using these TNBC cell lines with stable molecular subtypes.

## Introduction

The unfavorable prognosis of triple-negative breast cancer (TNBC) stems in part from a lack of effective targeted therapies and heterogeneity in clinical response to standard chemotherapy

**Data Availability Statement:** All relevant data are within the paper and its Supporting Information files.

**Funding:** This study was supported by: This study was supported by the Morgan Welch Inflammatory Breast Cancer Research Program, a State of Texas Rare and Aggressive Breast Cancer Research Program grant (N.T. Ueno), and MD Anderson's Cancer Center Support Grant (P30CA016672, used the Animal Core Facility). Insight Genetics Inc. provided support for this study in the form of salaries for RSS, DRH, BLS, TJN, ORL. The specific roles of these authors are articulated in the 'author contributions' section. All funders in this study provided support in the form of research materials, but did not have any additional role in the study design, data collection and analysis, decision to publish, or preparation of the manuscript.

**Competing interests:** The authors of this paper have read the journal's policy and the authors of this paper have the following competing interests: NTU has contracted research with Insight Genetics Inc. (http://www.insightgenetics.com/). RSS, DRH, BLS, TJN, ORL are all paid employees of Insight Genetics, Inc. This does not alter our adherence to PLOS ONE policies on sharing data and materials. The authors would like to declare the following patents/patent applications associated with this research: Insight Genetics Inc. has taken an exclusive license to a patent application for the analysis method to categorize TNBCType patients into various subtypes. The patent application is as follows: [US14/358,330]. Gene expression datasets can be uploaded for classification by TNBCType at http://cbc.mc.vanderbilt.edu/tnbc/. TNBCType-IM is a proprietary algorithm of Insight Genetics Inc.

[1]. There is an urgent unmet need to identify features of TNBC that can predict response to current standard cytotoxic treatment and facilitate the development of new targeted therapies for this disease.

Towards filling this need, significant efforts have been made to define the molecular heterogeneity of TNBC and to correlate these molecular signatures with clinical outcome and therapeutic effectiveness. In a meta-analysis published in 2009, Lehmann *et al.* performed cluster expression analysis on 21 breast cancer datasets containing 587 TNBC cases and identified a set of 2,188 genes that could classify TNBC into six molecular subtypes displaying unique characteristics: basal-like 1 (BL1) and basal-like 2 (BL2), characterized by the presence of cell cycle and DNA damage response genes; immunomodulatory (IM); mesenchymal (M); mesenchymal stem-like (MSL); and luminal androgen receptor (LAR) [2]. The classification algorithm based on these 2,188 genes is referred to as TNBCtype.

Later, Lehman *et al.*, recognizing the variety in the histological features of the tumor specimens used to identify the TNBCtype set of genes and aiming to determine whether stromal elements contributed to the molecular classification of any subtype, refined this TNBC classification algorithm [3]. Using histopathological quantification and laser-capture microdissection, they determined that the IM and MSL subtypes were the result of infiltrating lymphocytes and tumor-associated stromal cells, respectively. Thus, the authors concluded that IM status should be determined independently of subtype, which led to removal of the IM and MSL subtypes and left a revised classification with 4 subtypes, BL1, BL2, M, and LAR [3], referred to as the TNBCtype-4 classification.

To overcome obstacles inherent to the original TNBCtype and TNBCtype-4 algorithms and thereby facilitate clinical adoption of TNBC subtyping and improve reproducibility, Ring *et al.* built a new 101-gene algorithm, TNBCtype-IM (Insight Genetics), using the same gene expression datasets used to develop the original TNBCtype algorithm (Table 1) [4]. Samples classified as IM using an independent model were removed, and shrunken centroids were used to define a minimal gene set for five subtypes: BL1, BL2, LAR, M, and MSL. The subtypes assigned by TNBCtype-IM matched the subtypes assigned by TNBCtype in 87% of cases in a set of seven TNBC cohorts and in 88% of cases in an independent cohort [4]. Evaluation of molecular subtype by principal component analysis revealed that IM could be recognized by the TNBCtype algorithm as a feature distinct from the intrinsic TNBC subtypes of BL1, BL2, LAR, and M; thus, the TNBCtype-IM algorithm determines IM status in addition to each subtype (for example, BL1/IM-positive or BL1/IM-negative) [5].

In preclinical and translational research, cell lines and xenograft models are frequently used to identify the biological and cellular properties of distinct breast cancer subtypes. Heterogeneity in molecular background between cell lines thought to belong to the same subtype can influence therapeutic response; this raises the question of which particular cell line should be chosen for study [6]. Many investigators have used TNBCtype to select cell lines for

**Table 1. Comparison of TNBC molecular subtypes assigned by TNBCtype and TNBCtype-IM algorithms.**

| TNBCtype subtype | Corresponding TNBCtype-IM subtype |
|---|---|
| Basal-like 1 | Basal-like 1 |
| Basal-like 2 | Basal-like 2 |
| Immunomodulatory | No corresponding subtype; IM is a potential modifier for each molecular subtype |
| Mesenchymal | Mesenchymal |
| Mesenchymal stem-like | Mesenchymal stem-like |
| Luminal androgen receptor | Luminal androgen receptor |
| Unstable | Unstable |

investigational research; however, as mentioned above, the original classification has been revised to include five subtypes and the IM modifier. We speculated that TNBC cell lines with concordant molecular subtypes per TNBCtype and TNBCtype-IM are the most representative of their molecular subtypes and should be preferred models for translational research. The purpose of the study reported here was to identify xenograft-transplantable TNBC cell lines that maintained their molecular definition between the two algorithms and between *in vitro* and *in vivo* analyses, so as to identify the most appropriate models for preclinical study.

## Materials and methods

### Cell lines and cell culture conditions

Seventeen human TNBC cell lines were used in this study as described in Table 2. The cell lines were purchased from American Type Culture Collection with the exception of SUM159PT and SUM149PT, which were purchased from Asterand Bioscience, and HCC3153, from The University of Texas Southwestern Medical Center. Cell lines were grown in a humidified sterile incubator at 37°C in an atmosphere of 5% $CO_2$. The MDAMB231, MDAMB468, MDAMB453, BT549, MDAMB157, DU4475, MDAMB436, and BT20 cell lines were maintained in DMEM/

**Table 2. Molecular subtypes of 17 TNBC cell lines and xenografts derived from the same cell lines according to classification with the TNBCtype-IM algorithm.**

| Cell Line | Subtype per TNBCtype cell line data | Subtype per TNBCtype-IM cell line data | Subtype per TNBCtype Xenograft model | Subtype per TNBCtype-IM Xenograft model |
|---|---|---|---|---|
| *Concordant Stable Subtypes* | | | | |
| HCC70 | BL2 (0.24) | BL2 (0.27)[1] | BL2 (0.36) | BL2 (0.38) |
| SUM149PT | BL2 (0.3) | BL2 (0.21)[2] | BL2 (0.40) | BL2 (0.37) |
| HCC1806 | BL2 (0.22) | BL2 (0.26) | BL2 (0.42) | BL2 (0.49) |
| BT549 | M (0.21) | M (0.15) | M (0.40) | M (0.41) |
| MDAMB453 | LAR (0.53) | LAR (0.4) | LAR (0.37) | LAR (0.38) |
| HCC2157 | BL1 (0.66) | BL1 (0.4) | BL1 (0.68) | BL1 (0.33) |
| *Discordant or Unstable Subtypes* | | | | |
| SUM185PE | LAR (0.39) | UNS | UNS | LAR (0.32) |
| BT20 | UNS | BL2 (0.18) | LAR (0.36) | LAR (0.32) |
| MDAMB157 | MSL (0.25) | LAR (0.12) | BL2 (0.21) | LAR (0.17) |
| SUM159PT | MSL (0.14) | BL2 (0.18)[3] | BL2 (0.47) | BL2 (0.54) |
| MDAMB468 | BL1 (0.19) | BL2 (0.2)[4] | UNS | BL2 (0.24)[5] |
| MDAMB231 | MSL (0.12) | BL2 (0.24) | UNS | BL2 (0.25) |
| HCC1187 | IM (0.22) | BL2 (0.17) | BL2 (0.32) | BL2 (0.17)[6] |
| DU4475 | IM (0.17) | BL1 (0.14) | UNS | UNS |
| MDAMB436 | MSL (0.13) | UNS | LAR (0.33) | LAR (0.39) |
| HCC1937 | BL1 (0.28) | BL2 (0.37) | BL1 (0.37) | BL2 (0.34)[7] |
| HCC3153 | BL1 (0.24) | BL1 (0.37) | UNS | M (0.45) |

The values in parentheses are correlation values.

[1] Dual subtype of BL1 (0.25)

[2] Dual subtype of M (0.17)

[3] Dual subtype of LAR (0.16)

[4] Dual subtype of BL1 (0.13)

[5] Dual subtype of M (0.23)

[6] Dual subtype of BL1 (0.14)

[7] Dual subtype of M (0.19)

F12 supplemented with fetal bovine serum (10%) and penicillin/streptomycin (100 U/mL). The HCC1187, HCC1806, HCC70, HCC1937, and HCC3153 cell lines were maintained in RPMI1640 supplemented with fetal bovine serum (10%) and penicillin/streptomycin (100 U/mL). The SUM159PT, SUM149PT, and SUM185PE cell lines were maintained in F12 medium supplemented with insulin (5 μg/mL) and hydrocortisone (1 μg/mL). Cell lines were validated using a short-tandem-repeat method based on a primer extension to detect single-base derivations by the Characterized Cell Line Core Facility at The University of Texas MD Anderson Cancer Center.

## Establishment of xenograft tumors

Tumor xenografts from 17 TNBC cell lines (Table 2) were analyzed in this study. All animal experiments were approved by the Institutional Animal Care and Use Committee (protocol 1305-RN01) of MD Anderson Cancer Center. Human xenograft tumors were established in 4- to 6-week-old immunocompromised mice (Nod-SCID-Gamma) that were bred in-house (Department of Experimental Radiation Oncology, MD Anderson Cancer Center) and housed in pathogen-free conditions within the MD Anderson Research Animal Support Facility. Mice were treated in accordance with NIH guidelines and received standard chow and water *ad libitum*. Individual tumor xenografts were established in anesthetized mice by implanting $5 \times 10^6$ TNBC cells, re-suspended in a 50:50 Matrigel:PBS solution, into the fourth inguinal mammary gland. Established tumors were monitored three times weekly by caliper measurements, and mice were euthanized by $CO_2$ asphyxiation when tumors reached 750 mm$^3$. Excised tumors were fixed in 10% buffered formalin and embedded in paraffin. Tumor blocks were sectioned (5 μm thick) and mounted onto poly-L-lysine glass slides. Five slides for each tumor were prepared histologically, whole-scraped, and processed collectively using QIAGEN's RNeasy FFPE Kit (Hilden, Germany) according to the manufacturer's recommendations.

## Subtyping of TNBC cell lines and xenografts

Normalized data from the GSE-10890 and E-TABM-157 publicly available gene data sets were used to classify the 17 TNBC cell lines using the original 2,188-gene algorithm (TNBCtype) and RPKM expression data provided by the Cancer Cell Line Encyclopedia (DepMap Public 19Q3), or GSE-10890 in the case of HCC3153, for the 101-gene TNBCtype-IM algorithm [2,4,7]. We were not able to use E-TABM-157 gene data for analysis with the modified 101-gene TNBCtype-IM algorithm because of a significant number (greater than 10%) of missing genes.

Gene expression profiles were created for the TNBC xenograft tumors by using exome capture-based RNA sequencing on RNA samples derived from respective tumors. Briefly, exome-enriched cDNA libraries were constructed using TruSeq RNA Exome (Illumina, San Diego, CA) according to the manufacturer's recommendations. Libraries were loaded on a NextSeq 500 sequencing system (Illumina, San Diego, CA) with a high-output v3 150 cycle reagent kit, with a mean of 25 million reads per sample. Base call files from each sequencing run were converted to fastq format using bcl2fastq conversion software (Illumina, San Diego, CA) and aligned to the Ensembl GRCh37 *Homo sapiens* reference using STAR (Spliced Transcripts Alignment to a Reference, v.020201). Transcript assembly and expression analysis were performed on each sample with cufflinks v. 2.2.1, resulting in fragments per kilobase million (FPKM) values for each transcript in the genes of interest, which were summed into one FPKM value for each gene [8,9]. The resulting FPKM data for each sample were compiled into a comma-separated values file and analyzed using the original TNBCtype and TNBCtype-IM algorithms to establish the subtype and determine whether IM features were present.

Gene expression profiles from the cell lines were correlated to the centroids for each of the TNBC subtypes defined in each algorithm using Spearman's test, because we have observed gene expression profiles to change in a monotonic relationship but not necessarily in a linear relationship between the various subtypes, and this is often best correlated using Spearman's test. Cell lines were assigned to the TNBC subtype with the highest correlation. Data were log base 2 transformed (either from FPKM or Affy), and then each gene was centered across the batch that was being analyzed. Correlation values were then determined by using Spearman's rank order method against a centroid value set for each subtype. The cutoff for each subtype was 0.1 (for consistency between TNBCtype and TNBCtype-IM), and all model coefficients and cutoffs were determined using the 14 discovery data sets used in the original Lehmann et al. analysis [2] and were not altered afterwards [4]. If multiple subtypes exceeded a correlation value of 0.1, the z-score method was performed for significance between the subtypes above the cutoff. If there was not a significant difference, multiple subtypes were reported ranked by their level of correlation. If one subtype surpassed the mathematically determined cutoff value, the cell line was assigned to that subtype. If more than one subtype surpassed the cutoff value with a significant difference between the two, the cell line was assigned to the predominant subtype. If more than one subtype surpassed the cutoff value with no significant difference between the two, the cell line was considered to have a dual subtype. For this study, subtype determinations were based on the highest correlation value to establish concordance between the two subtyping algorithms. Finally, if no subtype surpassed the cutoff value, the cell line was classified as unstable (UNS).

## Results

### TNBC cell lines display partial concordance between TNBCtype and TNBCtype-IM subtyping *in vitro*

Of the 17 *in vitro* TNBC cell lines evaluated, 41% (7/17) were classified similarly by the TNBCtype and TNBCtype-IM algorithms (HCC70, SUM149PT, HCC1806, BT549, MDAMB453, HCC2157, and HCC3153) (Table 2). The subtype on TNBCtype-IM was the same as the subtype on TNBCtype for 50% (1/2) of the cell lines classified as LAR by TNBCtype, 50% (2/4) of the cell lines classified as BL1 by TNBCtype, 100% (1/1) of the cell lines classified as M by TNBCtype, 100% (3/3) of the cell lines classified as BL2 by TNBCtype, and 0% (0/4) of the cell lines classified as MSL by TNBCtype. Two cell lines were classified as IM by TNBCtype but were not classified as IM by TNBCtype-IM, which suggests that this modifier requires a microenvironment with the presence of stromal cell infiltrates, which are lacking within *in vitro* culture. These findings seemed to support the hypothesis from Lehman *et al.* [3] that IM and MSL are not subtypes of TNBC but have additional underlying biology. Further, we tried to classify each cell line according to the subtype with the highest correlation, but four cell lines (HCC70, SUM149PT, SUM159PT, and MDAMB468) were classified as having dual subtypes on the basis of analysis of *in vitro* cell lines using the TNBCtype-IM algorithm. For comparisons between subtyping algorithms, we listed the subtype with the highest correlation in Table 2, but we also indicated the dual subtypes in footnotes to the table.

### Identification of six cell lines that display similar results in cell lines and animal models

Concordance was observed between the *in vitro* cell line subtype and the *in vivo* xenograft subtype in six of the 17 tumor xenograft models tested (HCC70, SUM149PT, HCC1806, BT549, MDAMB453 and HCC2157) (Table 2). To confirm the reproducibility of our results we

reclassified each of these six cell lines using the TNBCtype-IM algorithm and cell line data from 5 different sources and found that they were consistently classified to the same subtype (Table 3)(S1 Table). Similar to the *in vitro* assessment, no positive IM modifier or MSL subtype was detected in any of the xenograft tumor samples analyzed.

## Discussion

This is the first study to clearly define molecularly stable cell lines to represent the BL1, BL2, LAR, and M TNBC subtypes. We found that of the cell lines examined, HCC70 (BL2), SUM149PT (BL2), HCC1806 (BL2), BT549 (M), MDAMB453 (LAR) and HCC2157(BL1) were stable across both algorithms, between the *in vitro* and *in vivo* xenograft models and were consistently classified to the same subtype using multiple datasets, which demonstrates reproducibility (Table 3). Therefore, we consider these cell lines the most suitable representatives of their respective subtypes.

Previous work focusing on breast cancer cell line characterization has shown the difficulty of clearly determining molecular subtypes. A study that determined ER, PR, and HER2 expression in human breast cancer cell lines using immunohistochemical and immunocytochemistry assays did not find complete concordance between molecular subtypes determined using the two methods [10]. Another study determined protein expression using immunoblot analyses to characterize breast cancer cell lines, including 18 TNBC cell lines. Within each subtype, a significant level of genetic heterogeneity was found. Profiled pathway activation status was examined to determine activated pathways resulting from these mutational combinations, which provided better insight into the molecular classification of these cell lines [6].

Our study utilized two distinct gene-based algorithms to molecularly characterize TNBC cell lines *in vitro* and validated the results using *in vivo* animal tumor models derived from these cell lines, which we believe can give insight into which are the most molecularly stable and representative of their respective subtype. We tried to classify each cell line according to the subtype with the highest correlation, but four cell lines had confounding subtypes, most likely indicating dual subtypes that may express gene ontologies of more than one type. The molecular expression of more than one subtype can may call into question the suitability of these cell lines for research that may rely on the TNBCtype molecular classification. However, cell lines with dual subtypes may serve as models for studying the influence of external factors on the evolution of tumor subtype.

It is important to note that the IM subtype was not found in any of our specimens using the refined TNBCtype-IM algorithm, confirming the findings of Lehman *et al.* that IM should be considered an indicator of the presence of tumor-associated lymphocytes and determined

**Table 3. Subtypes of 6 molecularly stable TNBC cell lines and xenografts derived from multiple sources maintains subtype according to classification with the TNBCtype-IM algorithm.**

| Cell Line | Subtype per TNBCtype-IM from GSE15361 | Subtype per TNBCtype-IM from GSE10890 | Subtype per TNBCtype-IM from CCLE | Subtype per TNBCtype-IM Xenograft model | Subtype per TNBCtype Lehman, 2011 |
|---|---|---|---|---|---|
| HCC70 | BL2 (0.22) | BL2 (0.26) | BL2 (0.27) | BL2 (0.38) | BL2 (0.24) |
| SUM149PT | BL2 (0.14) | * | BL2 (0.21) | BL2 (0.37) | BL2 (0.30) |
| HCC1806 | * | BL2 (0.42) | BL2 (0.26) | BL2 (0.49) | BL2 (0.22) |
| BT549 | M (0.18) | M (0.11) | M (0.15) | M (0.41) | M (0.21) |
| MDAMB453 | LAR (0.30) | LAR (0.48) | LAR (0.40) | LAR (0.38) | LAR (0.53) |
| HCC2157 | BL1 (0.44) | * | BL1 (0.40) | BL1 (0.33) | BL1 (0.66) |

*Indicates cell line not represented in dataset.

independently of the subtype [3]. In fact, TNBCtype-IM was used in a recently published case report in which a patient eligible for immunotherapy tested negative for PD-L1 by immunohistochemistry but positive for IM by TNBCtype-IM. The patient had received exhaustive chemotherapy and experienced a complete radiologic response after four cycles of pembrolizumab [5]. Similar to what was observed in our analyses of cells *in vitro*, we were not able to identify positive IM status in the gene expression profiles of TNBC xenograft tumors. Since IM status has been shown to be dependent on the presence of tumor-infiltrating lymphocytes [3], it would be of interest to determine whether IM status could be observed in "humanized mice" bearing TNBC xenografts or freshly derived patient-derived xenograft models with human tumor-infiltrating immune cells.

Many published studies have adopted TNBCtype molecular subtyping for cell line selection for research. However, some cells lines have different molecular classifications in the *in vitro* and *in vivo* settings. Our results suggest that data interpretation and experimental planning should be interpreted with caution. Our data do not suggest that cell lines that demonstrate different molecular subtypes in the *in vitro* and *in vivo* settings are not suitable for experiments, but rather suggest that if the premise of the research is based on TNBCtype molecular classification, cell lines should be used that are molecularly stable regardless of the experimental condition.

## Conclusions

We identified several TNBC cell lines that have concordant molecular subtypes according to TNBCtype and TNBCtype-IM and between cell lines and xenografts. We believe that such cell lines are the most molecularly stable and the most representative of their respective subtype. In our study, those cell lines were HCC70 (BL2), SUM149PT (BL2), HCC1806 (BL2), BT549 (M), MDAMB453 (LAR) and HCC2157(BL1). Therefore, for drug development studies based on TNBCtype molecular subtyping, we recommend using these cell lines.

## Supporting information

**S1 Table. Correlation values from each subtype of the 6 molecularly stable TNBC cell lines and xenografts derived from multiple sources according to classification with the TNBCtype-IM algorithm.**
(DOCX)

## Acknowledgments

We thank Sunita Patterson and Stephanie Deming of the Department of Scientific Publications at MD Anderson Cancer Center for editing the manuscript, and Insight Genetics for the support of the gene expression data analysis.

## Author Contributions

**Conceptualization:** Jose Rodrigo Espinosa Fernandez, Bedrich L. Eckhardt, Jangsoon Lee, Bora Lim, Naoto T. Ueno.

**Data curation:** Jose Rodrigo Espinosa Fernandez, Bedrich L. Eckhardt, Jangsoon Lee, Rob S. Seitz, David R. Hout, Brock L. Schweitzer, Tyler J. Nielsen, O. Rayne Lawrence, Ying Wang, Arvind Rao.

**Investigation:** Jose Rodrigo Espinosa Fernandez, Bedrich L. Eckhardt, Jangsoon Lee, Troy Pearson.

**Methodology:** Bedrich L. Eckhardt, Jangsoon Lee, Troy Pearson, Rob S. Seitz, David R. Hout, Brock L. Schweitzer, Tyler J. Nielsen, O. Rayne Lawrence, Ying Wang, Arvind Rao.

**Software:** Jose Rodrigo Espinosa Fernandez, Rob S. Seitz, David R. Hout, Brock L. Schweitzer, Tyler J. Nielsen, O. Rayne Lawrence, Ying Wang, Arvind Rao.

**Supervision:** Naoto T. Ueno.

**Validation:** Jose Rodrigo Espinosa Fernandez, Bedrich L. Eckhardt, Jangsoon Lee, Rob S. Seitz, David R. Hout, Brock L. Schweitzer, Tyler J. Nielsen, O. Rayne Lawrence.

**Writing – original draft:** Jose Rodrigo Espinosa Fernandez.

**Writing – review & editing:** Jose Rodrigo Espinosa Fernandez, Bedrich L. Eckhardt, Jangsoon Lee, Bora Lim, Rob S. Seitz, David R. Hout, Brock L. Schweitzer, Tyler J. Nielsen, O. Rayne Lawrence, Ying Wang, Arvind Rao, Naoto T. Ueno.

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
