## [Decision Letter · Decision Letter 0]

21 Sep 2019

PONE-D-19-21744

Identification of triple-negative breast cancer cell lines classified under the same molecular subtype using different molecular characterization techniques: implications for translational research

PLOS ONE

Dear Dr. Ueno,

Thank you for submitting your manuscript to PLOS ONE. After careful consideration, we feel that it has merit but does not fully meet PLOS ONE’s publication criteria as it currently stands. Therefore, we invite you to submit a revised version of the manuscript that addresses the points raised during the review process.

The reviewers felt that this manuscript would be important in providing a framework for investigators to select in vitro and in vivo models of TNBC. Please address the reviewer comments, which focused on a lack of sufficient details in the classification strategy and which publicly available data were used for analysis and rationale for value cut-offs. In addition, please comment on how reproducible the classification calls are using independent public data sets. There were also several clarifications or improvements needed related to data presented in the Figures and data tables (Reviewers 1-2). Finally, a statistics reviewer raised questions related to correlation and cut-off values. 

We would appreciate receiving your revised manuscript by Nov 05 2019 11:59PM. To enhance the reproducibility of your results, we recommend that if applicable you deposit your laboratory protocols in protocols.io, where a protocol can be assigned its own identifier (DOI) such that it can be cited independently in the future. For instructions see: http://journals.plos.org/plosone/s/submission-guidelines#loc-laboratory-protocols

We look forward to receiving your revised manuscript.

Kind regards,

Tiffany N. Seagroves, PhD

Academic Editor

PLOS ONE

Journal Requirements:

2. At this time, we request that you  please report additional details in your Methods section regarding animal care, as per our editorial guidelines: 1) Please provide details of animal welfare (e.g., shelter, food, water, environmental enrichment) 2) please describe any steps taken to minimize animal suffering and distress, such as by administering anesthetics or analgesics, 3) please include the method of sacrifice and 4) Please describe the frequency of monitoring and the criteria used to assess animal health and well-being. Thank you for your attention to these requests.

3. We note you have included a table to which you do not refer in the text of your manuscript. Please ensure that you refer to Table 1 in your text; if accepted, production will need this reference to link the reader to the Table.

I have reas the Journal's Policy and the author of this manuscript have the following competing interests: NTU declares contracted research with Insight Genetics, Inc.. RSS, DRH, BLS, TJN, ORL are all employee of Insight Genetics.  The other authors declare no potential conflicts of interest.

We note that one or more of the authors are employed by a commercial company: Insight Genetics, Inc.

Reviewers' comments:

Reviewer's Responses to Questions

**Comments to the Author**

1. Is the manuscript technically sound, and do the data support the conclusions?

Reviewer #1: Yes

Reviewer #2: Partly

Reviewer #3: Partly

2. Has the statistical analysis been performed appropriately and rigorously? 

Reviewer #1: I Don't Know

Reviewer #2: I Don't Know

Reviewer #3: No

3. Have the authors made all data underlying the findings in their manuscript fully available?

Reviewer #1: No

Reviewer #2: No

Reviewer #3: Yes

4. Is the manuscript presented in an intelligible fashion and written in standard English?

Reviewer #1: Yes

Reviewer #2: Yes

Reviewer #3: Yes

5. Review Comments to the Author

Reviewer #1: PONE-D-19-21744

The manuscript by Fernandez and colleagues relays important information for the molecular classification of TNBC subtypes in cell line models. There are a number of points that should be clarified and/or expanded prior to publication.

1. The methods indicate that publicly available gene expression data were used to classify the 28 TNBC cell lines. The authors should explicitly state which publically available data. Two references were cited. How was the data from the different articles analyzed? Analysis of this data should be more thoroughly explained in the methods section. Additionally, could the use of publicly available data have an impact on discordant results between cell line and xenograft results? For discordant results did the authors repeat RNA sequencing from cell lines grown in their own laboratory and compare to their RNA sequencing analysis from the xenograft studies presented here?

2. Figure 1: could the authors please indicate which cell lines are represented in Figure 1. Also, please include the data for each cell line in a supplemental file.

3. In Table 1 it would be helpful to annotate which cell lines were classified as dual subtypes.

4. The authors should present the analysis using TNBCtype from the xenograft data in addition to the TNBCtype-IM (Table 3).

5. Page 9, line 177 indicates that 6 of 17 tumor xenograft models tested were concordant, but in Table 3 HCC70 cell line and xenograft data also appears to be concordant for BL2.

6. Please clarify, were xenografts only established from 17/28 because 11 of the cell lines did not grow in vivo? Also clarify, was xenograft tumor data from an n=1 for each cell line?

Reviewer #2: Identification of triple negative breast cancer cell lines classified under the same molecular subtype using different molecular characterization techniques: implications of translational research.

Fernandez et al.

Manuscript: PONE-D-19-21744

The study by Fernandez et al. investigates the classification of triple negative breast cancer cell lines grown in vitro and as orthotopic xenograft transplant models in vivo. The goal is to define the TNBC subtypes of these cell lines using the original TNBCtype and modified TNBCtype –IM algorithms. The impact of the proposed studies lies in defining the molecular subtypes in order to provide the scientific community a framework by which to select these lines, either in vitro or in vivo, to model TNBC subtypes for mechanistic and/or preclinical studies. While the current study does address this goal, I have several moderate and minor concerns that I have outlined below.

Major Concerns:

1. The classification strategy is not well described. The authors have not defined the correlation cut-off used to define the subgroup classification (line 154-157) nor have they described the rationale for the selection of that value. Likewise, the authors state (line 153-154): “If more than one subtype surpassed the cut-off value with no significant difference between the two...” It was not clear how the authors determined whether there was a significant difference. In general, the methods should be more clearly written.

2. The authors should provide a table of correlation coefficients for each cell lines and each subgroup. While the authors have made the calls for each cell line, it would allow the reader to gauge the strength of the correlation if these data were provided.

3. It is not clear how reproducible these classification calls are. The authors should use an independent dataset (i.e. CCLE RNAseq data) to examine subtype classification for these cell lines. This would provide additional confidence that subtype calls are not dependent on culturing methods, specific growth conditions or technical or experimental variables. This is particularly relevant as subtype calls for 8/28 or 28.6% of the cell lines analyzed in the current study and the original Lehman JCI paper are not concordant using the TNBCtype calls (Table 2 of this study vs. Table 3 in Lehman 2011 JCI paper).

Minor Concerns:

1. Figure 1 is not very clear. It also appears to be lacking a legend as well as y-axis labels.

2. Table 2 and 3 seem somewhat redundant and the data could be merged into a single table.

3. Reference 2 appear to be merged with Reference 1 in the References Cited section.

Reviewer #3: A) Spearman's correlation test was used to determine sub-type categories. A score of .195 was presented in Figure 1 as the cutoff selected; however, that score is considered "weak correlation" at best. Could the author address this as well as why Spearman's was selected over other tests available?

B) The number of available samples in each subcategory seems rather now. In MSL category only two of the available four was correctly selected.

6. PLOS authors have the option to publish the peer review history of their article (what does this mean?). If published, this will include your full peer review and any attached files.

Reviewer #1: No

Reviewer #2: No

Reviewer #3: No

---

## [Author Response · Author response to Decision Letter 0]

27 Dec 2019

Reviewers' comments:

Reviewer #1: PONE-D-19-21744

The manuscript by Fernandez and colleagues relays important information for the molecular classification of TNBC subtypes in cell line models. There are a number of points that should be clarified and/or expanded prior to publication.

1. The methods indicate that publicly available gene expression data were used to classify the 28 TNBC cell lines. The authors should explicitly state which publically available data. Two references were cited. How was the data from the different articles analyzed? Analysis of this data should be more thoroughly explained in the methods section. Additionally, could the use of publicly available data have an impact on discordant results between cell line and xenograft results? For discordant results did the authors repeat RNA sequencing from cell lines grown in their own laboratory and compare to their RNA sequencing analysis from the xenograft studies presented here?

We appreciate the reviewer’s comment. Normalized data from the CCLE (RNAseq), GSE-10890, and E-TABM-157 publicly available gene data sets were used to classify the 17 TNBC cell lines using both the original 2,188-gene algorithm (TNBCtype) and the 101-gene TNBCtype-IM algorithm. We added a reference for this data sets: Neve RM, Chin K, Fridlyand J, Yeh J, Baehner FL, Fevr T, et al. A collection of breast cancer cell lines for the study of functionally distinct cancer subtypes Cancer Cell. 2006; 10(6): 515–527, where the methods are explained. 

In terms of discordance, it was our intention to find TNBC cell lines that would remain stable with regard to molecular subtype regardless of the algorithm or type of model (in vitro, in vivo) used. Please also see our responses below to reviewer 2.

2. Figure 1: could the authors please indicate which cell lines are represented in Figure 1. Also, please include the data for each cell line in a supplemental file. 

To avoid confusion, we decided to eliminate Figure 1from the manuscript because it does not represent any particular cell line data. Also, the correlation values for each cell line have been added to Table 2. 

3. In Table 1 it would be helpful to annotate which cell lines were classified as dual subtypes.

We thank the reviewer for this comment. No cell line data results are described in Table 1. However, we have annotated dual subtypes in Table 2 and added correlation values.

4. The authors should present the analysis using TNBCtype from the xenograft data in addition to the TNBCtype-IM (Table 3).

We agree with the reviewer. We have added the results of TNBCtype analysis from the xenograft data to table 2 to further demonstrate molecular subtype stability.

5. Page 9, line 177 indicates that 6 of 17 tumor xenograft models tested were concordant, but in Table 3 HCC70 cell line and xenograft data also appears to be concordant for BL2.

We agree with the reviewer and have updated our discussion of the results in Table 2 accordingly.

6. Please clarify, were xenografts only established from 17/28 because 11 of the cell lines did not grow in vivo? Also clarify, was xenograft tumor data from an n=1 for each cell line? 

Only 17 xenografts were established because those are the animal models we had available in our research laboratory. To reduce confusion, we have eliminated all mentions of the 11 cell lines that were not used for xenograft tumor generation. The data from these cell lines (formerly in Table 2) are largely redundant with the xenograft table and data did not meet the bar set forth by this manuscript of demonstrating subtype stability. Various changes have been made in the manuscript as a result of our decision to not mention the 11 cell lines for which xenografts were not established.

Reviewer #2: Identification of triple negative breast cancer cell lines classified under the same molecular subtype using different molecular characterization techniques: implications of translational research.

Fernandez et al.

Manuscript: PONE-D-19-21744

The study by Fernandez et al. investigates the classification of triple negative breast cancer cell lines grown in vitro and as orthotopic xenograft transplant models in vivo. The goal is to define the TNBC subtypes of these cell lines using the original TNBCtype and modified TNBCtype –IM algorithms. The impact of the proposed studies lies in defining the molecular subtypes in order to provide the scientific community a framework by which to select these lines, either in vitro or in vivo, to model TNBC subtypes for mechanistic and/or preclinical studies. While the current study does address this goal, I have several moderate and minor concerns that I have outlined below.

Major Concerns:

1. The classification strategy is not well described. The authors have not defined the correlation cut-off used to define the subgroup classification (line 154-157) nor have they described the rationale for the selection of that value. Likewise, the authors state (line 153-154): “If more than one subtype surpassed the cut-off value with no significant difference between the two...” It was not clear how the authors determined whether there was a significant difference. In general, the methods should be more clearly written.

We appreciate these comments. To address them, we have added the following to our methods section:

Data was log base 2 transformed (either from RPKM or Affy), followed by centering of each gene across the batch that was being analyzed. Correlation values were then determined by using Spearman’s rank order method against a centroid value set for each subtype. The cutoff for each subtype was 0.1 (for consistency between TNBCtype and TNBCtype-IM), all model coefficients and cutoffs were determined using the 14 discovery data sets, as in the original Lehmann et al. analysis [2], and were not altered afterwards [4]. If multiple subtypes exceed a correlation value of 0.1, the z-score method was performed for significance between the subtypes above the cutoff. If there was not a significant difference, then multiple subtypes were reported ranked by their level of correlation. (lines 154 to 163)

To better compare TNBCtype and TNBCtype-IM, we applied the correlation cutoff of 0.1 to TNBCtype-IM to match TNBCtype. This resulted in two xenograft results changing. Please see Table 2.

2. The authors should provide a table of correlation coefficients for each cell lines and each subgroup. While the authors have made the calls for each cell line, it would allow the reader to gauge the strength of the correlation if these data were provided.

We agree with the reviewer and have added these details to Table 2.

3. It is not clear how reproducible these classification calls are. The authors should use an independent dataset (i.e. CCLE RNAseq data) to examine subtype classification for these cell lines. This would provide additional confidence that subtype calls are not dependent on culturing methods, specific growth conditions or technical or experimental variables. This is particularly relevant as subtype calls for 8/28 or 28.6% of the cell lines analyzed in the current study and the original Lehman JCI paper are not concordant using the TNBCtype calls (Table 2 of this study vs. Table 3 in Lehman 2011 JCI paper).

We agree with the reviewer. We have re-analyzed all cell line data using the CCLE RNAseq data have generated the in vitro TNBCtype-IM subtypes (updated Table 2). The xenograft samples were analyzed using RNAseq, so this is a more appropriate comparison. In addition, we were not able to use the E-TABM-157 dataset used by Lehman due to several missing genes from the modified 101-gene algorithm (TNBCtype-IM). However, owing to availability, we used GSE-10890 to determine the subtype call for HCC3153 (see lines 130 to 135 and lines 181 to 195)

We do expect discordant calls between the two algorithms for the following reasons: 

1. As shown in the Ring paper, using a large dataset, we expect approximately 87% concordance between the two algorithms. 

2. TNBCtype-IM allows for a dual subtype call with the IM subtype removed and is included as a modifier for each subtype.

3. Using data previously reported, with histopathological quantification and laser-capture microdissection, it was determined that the IM and MSL subtypes as reported in the original TNBCtype algorithm were likely due to tumor-infiltrating lymphocytes and tumor-associated stromal cells, respectively. Therefore, we are more confident in the calls of TNBCtype-IM, which has been modified and does not bias cohorts towards finding samples of these two subtypes.

Minor Concerns:

1. Figure 1 is not very clear. It also appears to be lacking a legend as well as y-axis labels.

We agree with this comment. We have eliminated this figure and added details to the Methods section. 

2. Table 2 and 3 seem somewhat redundant and the data could be merged into a single table.

We agree with the reviewer and have made this modification. See lines 34 to 42.

3. Reference 2 appear to be merged with Reference 1 in the References Cited section.

Thank you for the comment. We have corrected this error. 

Reviewer #3: A) Spearman's correlation test was used to determine sub-type categories. A score of .195 was presented in Figure 1 as the cutoff selected; however, that score is considered "weak correlation" at best. Could the author address this as well as why Spearman's was selected over other tests available?

The correlation value cutoffs selected have been determined empirically. We have observed gene expression profiles to change in a monotonic but not necessarily linear relationship between the various subtypes, which is often best correlated using Spearman’s test as compared to Pearson’s test. We have added this information to the Methods section. (lines 150 to 153)

B) The number of available samples in each subcategory seems rather now. In MSL category only two of the available four was correctly selected.

Using data previously reported, with histopathological quantification and laser-capture microdissection, it was determined that the IM and MSL subtypes as reported in the original TNBCtype algorithm were likely due to infiltrating lymphocytes and tumor-associated stromal cells, respectively. It is because of these data, in combination with the new centering method, that it is possible the MSL subtype was overrepresented with the original TNBCtype algorithm and is more accurately represented using TNBCtype-IM.

---

## [Decision Letter · Decision Letter 1]

22 Jan 2020

PONE-D-19-21744R1

Identification of triple-negative breast cancer cell lines classified under the same molecular subtype using different molecular characterization techniques: implications for translational research

PLOS ONE

Dear Dr. Ueno,

Thank you for submitting your manuscript to PLOS ONE. After careful consideration, we feel that it has merit but does not fully meet PLOS ONE’s publication criteria as it currently stands. Therefore, we invite you to submit a revised version of the manuscript that addresses the points raised during the review process.

Please address Reviewer #2's concerns about rigor and reproducibility of the data sets, including concerns about correlation coefficients and variance that may be introduced by different growth conditions among laboratories. In particular, please address this comment: "..the investigators should demonstrate that the same cell line, grown in the same way (i.e. in vitro or in vivo) is consistently classified to the same subtype; there are multiple publicly available RNAseq datasets that can be used to complete these studies." It would also be useful to address some of the reviewers concerns and potential limitations of the study in the discussion, which Reviewer #2 found to be brief.

We would appreciate receiving your revised manuscript by Mar 07 2020 11:59PM. To enhance the reproducibility of your results, we recommend that if applicable you deposit your laboratory protocols in protocols.io, where a protocol can be assigned its own identifier (DOI) such that it can be cited independently in the future. For instructions see: http://journals.plos.org/plosone/s/submission-guidelines#loc-laboratory-protocols

We look forward to receiving your revised manuscript.

Kind regards,

Tiffany N. Seagroves, Ph.D.

Academic Editor

PLOS ONE

Reviewers' comments:

Reviewer's Responses to Questions

**Comments to the Author**

1. If the authors have adequately addressed your comments raised in a previous round of review and you feel that this manuscript is now acceptable for publication, you may indicate that here to bypass the “Comments to the Author” section, enter your conflict of interest statement in the “Confidential to Editor” section, and submit your "Accept" recommendation.

Reviewer #1: All comments have been addressed

Reviewer #2: (No Response)

Reviewer #3: All comments have been addressed

2. Is the manuscript technically sound, and do the data support the conclusions?

Reviewer #1: Yes

Reviewer #2: Partly

Reviewer #3: Yes

3. Has the statistical analysis been performed appropriately and rigorously? 

Reviewer #1: Yes

Reviewer #2: Yes

Reviewer #3: Yes

4. Have the authors made all data underlying the findings in their manuscript fully available?

Reviewer #1: Yes

Reviewer #2: No

Reviewer #3: Yes

5. Is the manuscript presented in an intelligible fashion and written in standard English?

Reviewer #1: Yes

Reviewer #2: Yes

Reviewer #3: Yes

6. Review Comments to the Author

Reviewer #1: (No Response)

Reviewer #2: The revised manuscript is improved; however a few concerns remain. First, the level of rigor in the subtype calls is not clear. The authors should report the correlation coefficient values for each cell line or xenograft and each TNBC subtype. Given that most of the reported correlation coefficients are low, this would afford the reader additional insight into the strength of the calls and potential concerns with selecting any given model system for subsequent studies. Secondly, while the authors do demonstrate concordance in subtype calls when specific cell lines are grown in vitro or as an in vivo xenograft, it is unclear how reproducible the subtypes call is between multiple datasets. As I noted in my previous review, and in this review, the subtype correlation coffecicients are relatively weak. As such, the investigators should demonstrate that the same cell line, grown in the same way (i.e. in vitro or in vivo) is consistently classified to the same subtype; there are multiple publicly available RNAseq datasets that can be used to complete these studies. If this is not reproducible, there are concerns with the selection of these lines for future studies as growth conditions will undoubtedly vary between laboratories. Finally, the manuscript is relatively well written, but there are a number of awkwardly worded sections and the discussion is rather brief.

Reviewer #3: This reviewer believes the authors have completed all changes needed in the manuscript. The paper should be accepted.

7. PLOS authors have the option to publish the peer review history of their article (what does this mean?). If published, this will include your full peer review and any attached files.

Reviewer #1: No

Reviewer #2: No

Reviewer #3: No

---

## [Author Response · Author response to Decision Letter 1]

9 Mar 2020

Dear Dr. Heber,

Thank you for the opportunity to revise our manuscript “Identification of triple-negative breast cancer cell lines classified under the same molecular subtype using different molecular characterization techniques: implications for translational research.” We have addressed the reviewers’ concerns, as indicated by our point-by-point responses to the comments in italics below.

Reviewers' comments:

Reviewer #2: The revised manuscript is improved; however a few concerns remain. First, the level of rigor in the subtype calls is not clear. The authors should report the correlation coefficient values for each cell line or xenograft and each TNBC subtype. Given that most of the reported correlation coefficients are low, this would afford the reader additional insight into the strength of the calls and potential concerns with selecting any given model system for subsequent studies. Secondly, while the authors do demonstrate concordance in subtype calls when specific cell lines are grown in vitro or as an in vivo xenograft, it is unclear how reproducible the subtypes call is between multiple datasets. As I noted in my previous review, and in this review, the subtype correlation coffecicients are relatively weak. As such, the investigators should demonstrate that the same cell line, grown in the same way (i.e. in vitro or in vivo) is consistently classified to the same subtype; there are multiple publicly available RNAseq datasets that can be used to complete these studies. If this is not reproducible, there are concerns with the selection of these lines for future studies as growth conditions will undoubtedly vary between laboratories. Finally, the manuscript is relatively well written, but there are a number of awkwardly worded sections and the discussion is rather brief.

We thank the reviewer for the suggestions. First, we are aware that the correlation coefficient values we are reporting are low so in order to offer the reader additional insight into the strength of each subtype call, we have added a supplementary table where we report correlation values from each

subtype of the 6 molecularly stable TNBC cell lines and xenografts derived from multiple sources according to classification with the TNBCtype-IM algorithm.

Secondly, to demonstrate the strength and reproducibility of our results despite the low correlation coefficient values, we reproduced the calls for the six specific concordant cell lines (SUM149PT, HCC1806, SUM149PT, BT549, MDA-MB-453, and HCC2157) using 5 different datasets (GSE15361, GSE10890, CCLE, Xenograft model, and Lehman, 2011). We found that all cell lines were consistently classified to the same subtype across all datasets. We have included this data in the results and discussion section, and we have also added the results and source of each dataset in table 3.

Sincerely,

Naoto T. Ueno, M.D., Ph.D., F.A.C.P.

Professor of Medicine

---

## [Decision Letter · Decision Letter 2]

6 Apr 2020

Identification of triple-negative breast cancer cell lines classified under the same molecular subtype using different molecular characterization techniques: implications for translational research

PONE-D-19-21744R2

Dear Dr. Ueno,

We are pleased to inform you that your manuscript has been judged scientifically suitable for publication and will be formally accepted for publication once it complies with all outstanding technical requirements.

With kind regards,

Tiffany Seagroves

Academic Editor

PLOS ONE

Additional Editor Comments (optional):

Reviewers' comments:

Reviewer's Responses to Questions

**Comments to the Author**

1. If the authors have adequately addressed your comments raised in a previous round of review and you feel that this manuscript is now acceptable for publication, you may indicate that here to bypass the “Comments to the Author” section, enter your conflict of interest statement in the “Confidential to Editor” section, and submit your "Accept" recommendation.

Reviewer #2: All comments have been addressed

2. Is the manuscript technically sound, and do the data support the conclusions?

Reviewer #2: Yes

3. Has the statistical analysis been performed appropriately and rigorously? 

Reviewer #2: Yes

4. Have the authors made all data underlying the findings in their manuscript fully available?

Reviewer #2: Yes

5. Is the manuscript presented in an intelligible fashion and written in standard English?

Reviewer #2: Yes

6. Review Comments to the Author

Reviewer #2: The authors have adequately addressed my concerns.

7. PLOS authors have the option to publish the peer review history of their article (what does this mean?). If published, this will include your full peer review and any attached files.

Reviewer #2: No

---

## [Editor Report · Acceptance letter]

21 Apr 2020

PONE-D-19-21744R2 

Identification of triple-negative breast cancer cell lines classified under the same molecular subtype using different molecular characterization techniques: implications for translational research 

Dear Dr. Ueno:

I am pleased to inform you that your manuscript has been deemed suitable for publication in PLOS ONE. Congratulations! Your manuscript is now with our production department. 

With kind regards,

on behalf of

Dr. Tiffany Seagroves 

Academic Editor

PLOS ONE